# Stable transplantation of human mitochondrial DNA by high-throughput, pressurized isolated mitochondrial delivery

Alexander J Sercel[1†], Alexander N Patananan[2†], Tianxing Man[3], Ting-Hsiang Wu[4,5], Amy K Yu[1], Garret W Guyot[2], Shahrooz Rabizadeh[4,5,6,7], Kayvan R Niazi[4,5,7,8], Pei-Yu Chiou[3,7,8], Michael A Teitell[1,2,7,8,9,10,11]*

[1]Molecular Biology Interdepartmental Doctoral Program, University of California, Los Angeles, Los Angeles, United States; [2]Department of Pathology and Laboratory Medicine, David Geffen School of Medicine, University of California, Los Angeles, Los Angeles, United States; [3]Department of Mechanical and Aerospace Engineering, University of California, Los Angeles, Los Angeles, United States; [4]NanoCav, LLC, Culver City, United States; [5]NantBio, Inc, and ImmunityBio, Inc, Culver City, United States; [6]NantOmics, LLC, Culver City, United States; [7]California NanoSystems Institute, University of California, Los Angeles, Los Angeles, United States; [8]Department of Bioengineering, University of California, Los Angeles, Los Angeles, United States; [9]Eli and Edythe Broad Center of Regenerative Medicine and Stem Cell Research University of California, Los Angeles, Los Angeles, United States; [10]Department of Pediatrics, David Geffen School of Medicine, University of California, Los Angeles, Los Angeles, United States; [11]Jonsson Comprehensive Cancer Center, David Geffen School of Medicine, University of California, Los Angeles, Los Angeles, United States

*For correspondence:
mteitell@mednet.ucla.edu

[†]These authors contributed equally to this work

**Abstract** Generating mammalian cells with specific mitochondrial DNA (mtDNA)–nuclear DNA (nDNA) combinations is desirable but difficult to achieve and would be enabling for studies of mitochondrial-nuclear communication and coordination in controlling cell fates and functions. We developed 'MitoPunch', a pressure-driven mitochondrial transfer device, to deliver isolated mitochondria into numerous target mammalian cells simultaneously. MitoPunch and MitoCeption, a previously described force-based mitochondrial transfer approach, both yield stable isolated mitochondrial recipient (SIMR) cells that permanently retain exogenous mtDNA, whereas coincubation of mitochondria with cells does not yield SIMR cells. Although a typical MitoPunch or MitoCeption delivery results in dozens of immortalized SIMR clones with restored oxidative phosphorylation, only MitoPunch can produce replication-limited, non-immortal human SIMR clones. The MitoPunch device is versatile, inexpensive to assemble, and easy to use for engineering mtDNA–nDNA combinations to enable fundamental studies and potential translational applications.

## Introduction

Mitochondrial DNA (mtDNA) and nuclear DNA (nDNA) genome coordination regulates metabolism, epigenome modifications, and other processes vital for mammalian cell survival and activity (*Patananan et al., 2018*; *Ryan and Hoogenraad, 2007*; *Singh et al., 2017*). Together, these

**eLife digest** Mitochondria are specialized structures within cells that generate vital energy and biological building blocks. Mitochondria have a double membrane and contain many copies of their own circular DNA (mitochondrial DNA), which include the blueprints to create just thirteen essential mitochondrial proteins.

Like all genetic material, mitochondrial DNA can become damaged or mutated, and these changes can be passed on to offspring. Some of these alterations are linked to severe and debilitating diseases. Both the double membrane of the mitochondria and their high number of DNA copies make treating such diseases difficult. A successful therapy must be capable of correcting almost every copy of mitochondrial DNA. However, the multiple copies of mitochondrial DNA create a problem for genetic research as current techniques are unable to reliably introduce particular mitochondrial mutations to all types of human cells to investigate how they may alter cell function.

Sercel, Patananan et al. have developed a method to deliver new mitochondria into thousands of cells at the same time. This technique, called MitoPunch, uses a pressure-driven device to propel mitochondria taken from donor cells into recipient cells without mitochondrial DNA to reestablish their function. Using human cancer cells and healthy skin cells that lack mitochondrial DNA, Sercel, Patananan et al. showed that cells that received mitochondria retained the new mitochondrial DNA. The technique uses readily accessible parts, meaning it can be performed quickly and inexpensively in any laboratory. It further only requires a small amount of donor starting material, meaning that even precious samples with limited material could be used as mitochondrial donors.

This new technique has several important potential applications for mitochondrial DNA research. It could be used in the lab to create large numbers of cell lineswith known mutations in the mitochondrial DNA to establish new systems that test drugs or probe the interaction between mitochondrial and nuclear DNA. It could be used to study a broad spectrum of biological questions since mitochondrial function is essential for several processes required for life. Critically, it could also be used as a starting point to develop next-generation therapies capable of treating inherited mitochondrial genetic diseases in severely affected patients.

genomes encode >1100 mitochondrial proteins, with only 13 essential electron transport chain (ETC) proteins encoded within the mtDNA (*Calvo and Mootha, 2010*). The mitochondrial proteins encoded in the mtDNA and the nDNA must be compatible to support mitochondrial ETC activity. Mutations in mtDNA can impair the ETC by altering nDNA co-evolved ETC complex protein interactions, causing defective cellular respiration and debilitating diseases (*Greaves et al., 2012*). Furthermore, the coordination of these two genomes to transcribe, translate, and potentially modify appropriate levels of their respective gene products to maintain energetic and metabolic homeostasis is essential to the proper functioning of the ETC (*Wolff et al., 2014*). As a result, methods that enable pairing of specific mtDNA and nDNA genotypes in tractable systems are key to understanding the basic biology of mitonuclear interactions and their implications for health and disease.

Our current inability to edit mtDNA sequences is a roadblock for many studies and potential applications. For example, endonucleases targeted to the mitochondrion inefficiently eliminate and cannot alter mtDNA sequences (*Bacman et al., 2018*). An exciting new bacterial cytidine deaminase toxin generates a limited repertoire of point mutations in the mtDNA; however, its efficiency remains low and it is unable to knock-in new gene sequences (*Mok et al., 2020*). Mitochondrial transfer between cells in vitro and in vivo provides a potential path forward for transplanting existing mtDNA sequences; however, the mechanisms controlling such transfers remain unknown (*Dong et al., 2017*; *Torralba et al., 2016*). Isolated mitochondrial transfer has been used to deliver mitochondria to a range of recipient cell types in vitro and even in vivo (*Caicedo et al., 2015*; *Emani et al., 2017*; *Kitani et al., 2014*); however, most studies using these methods observe only short-term changes to cell or organ performance and function. A small number of these studies have coincubated mitochondria with recipient cells and observed permanent retention of the exogenous mtDNA in mtDNA-deficient (so-called 'ρ0') cells using large doses of mitochondria or antibiotic selection

schemes (*Clark and Shay, 1982*; *Patel et al., 2017*), although these approaches may not be possible when mitochondrial donor material is limited or does not possess a suitable selection marker.

Methods that deliver mitochondria directly into ρ0 cells can increase stable mitochondrial transfer efficiency and employ a wider range of mitochondrial donor sources. Such methods include membrane disruption (*King and Attardi, 1988*; *Wu et al., 2016*) or fusion with enucleated cytoplasts (*Wilkins et al., 2014*). However, these methods are typically laborious, low-throughput, or depend on cancerous, immortal recipient cells lacking physiologic mitochondrial activity. An interesting recent study did report one desired mtDNA–nDNA clone and 11 false-positive clones using cybrid fusion with replication-limited cells, an achievement hampered by a low generation rate with unknown reproducibility or generalizability (*Wong et al., 2017*).

There exist clinically relevant methods to replace the mtDNA of human cells, such as somatic cell nuclear transfer and pronuclear transfer that involve delivering nuclear genetic material from patients with mtDNA diseases into enucleated oocytes with non-mutant mtDNA genotypes (*Hyslop et al., 2016*; *Tachibana et al., 2013*). These methods hold potential for replacing deleterious mtDNA for the unborn, but they are technically challenging, low-throughput, dependent on high-quality patient samples, and prone to contamination by mutant mtDNA from the affected nuclear source material (*Kang et al., 2016*). Higher-throughput techniques that exchange non-native for resident mtDNAs in non-immortal somatic cells in tissue culture could enable studies of mtDNA–nDNA interactions and replace deleterious mtDNAs within cells with therapeutic potential (*Patananan et al., 2016*). Thus, a higher throughput, reproducible, and versatile mtDNA transfer approach to generate multiple desired 'stable isolated mitochondrial recipient' (SIMR) clones in replication-limited cells remains essential for statistically valid studies and potential translation of mitochondrial transplantation.

## Results

### MitoPunch mechanism uses fluid pressure to disrupt the plasma membrane

We developed 'MitoPunch' as a simple, high-throughput mitochondrial transfer device consisting of a lower polydimethylsiloxane (PDMS) reservoir loaded with a suspension of isolated mitochondria, covered by a polyethylene terephthalate (PET) filter seeded with ~$1 \times 10^5$ adherent cells (*Figure 1A*, *Figure 1—figure supplement 1*). MitoPunch uses a solenoid-activated plunger to transfer isolated mitochondria in a holding chamber by force into the cytosol of mammalian cells. Upon actuation, a mechanical plunger deforms the PDMS from below, which, as calculated by numerical simulation, generates pressure up to 28 kPa inside the PDMS chamber, propelling the suspension through numerous 3 µm pores in the PET filter. This pressure cuts the plasma membrane of recipient cells sitting atop the pores and delivers mitochondria into the cytoplasm of the cut cells (*Figure 1B*). To assess performance, we compared MitoPunch to mitochondrial coincubation (*Kitani et al., 2014*) and to MitoCeption (*Caicedo et al., 2015*), a method that uses centripetal force generated in a centrifuge to localize mitochondria to recipient mammalian cells (*Figure 1C*). In MitoCeption, a 1500 × *g* centripetal force draws isolated mitochondria to a recipient cell monolayer. We calculate that the suspended mitochondria exert a pressure of ~1.6 Pa on recipient cell membranes (*Figure 1D*) (see Materials and methods).

### Mitochondrial delivery into transformed and primary cells

We isolated and delivered dsRed-labeled mitochondria from ~$1.5 \times 10^7$ HEK293T cells (*Miyata et al., 2014*) into ~$1 \times 10^5$ 143BTK– ρ0 osteosarcoma cells and replication-limited BJ ρ0 foreskin fibroblasts in technical triplicate and measured the fraction of recipient cells positive for dsRed fluorescence by ImageStreamx MarkII imaging flow cytometry (*Figure 2A*). We define technical replication as independently performed mitochondrial deliveries using the same isolated mitochondrial preparation into recipient cells of the same passage. For 143BTK– ρ0 cells at ~2 hr postdelivery, imaging flow cytometry showed that MitoPunch yielded the lowest fraction of dsRed-positive cells compared to coincubation or MitoCeption. Similarly, for BJ ρ0 recipient cells, MitoPunch yielded the lowest fraction of dsRed-positive cells compared to coincubation or MitoCeption, although at lower levels relative to 143BTK– ρ0 recipients. This measurement assesses colocalization of mitochondria with recipient cells, and not necessarily the occurrence or mechanism of

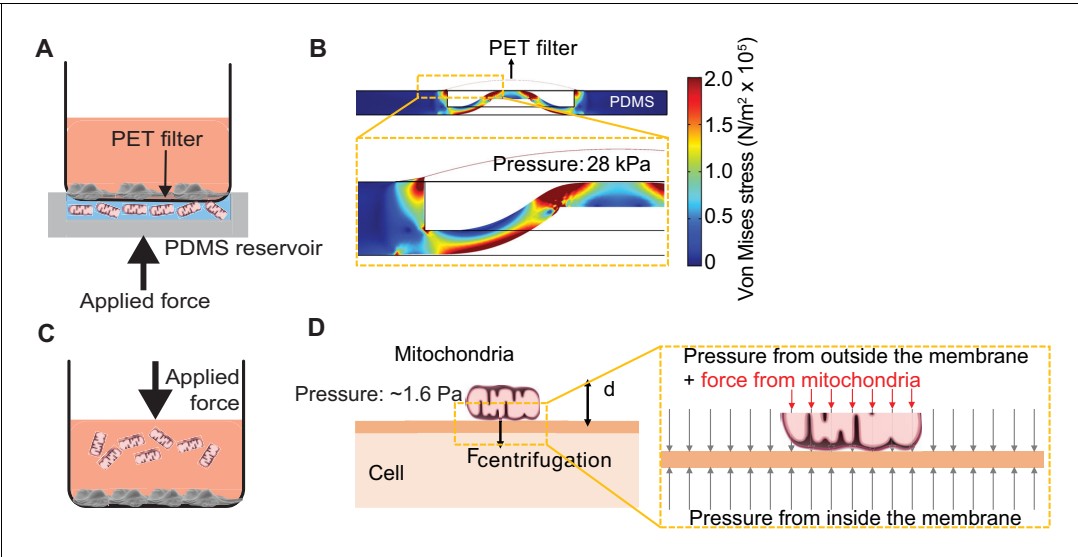

**Figure 1.** Pressure simulations of mitochondrial transfer tools. (**A**) Schematic of MitoPunch apparatus. Recipient cells ($1 \times 10^5$) are seeded on a porous polyester (PET) membrane ~24 hr before delivery. A freshly isolated suspension of mitochondria in 1× Dulbecco's Phosphate Buffered Saline (DPBS) with calcium and magnesium, pH 7.4, is loaded into the polydimethylsiloxane (PDMS) chamber and the filter insert is sealed over the PDMS before activation of the mechanical plunger to pressurize the apparatus and deliver the mitochondrial suspension into recipient cells. (**B**) Numerical simulation showing the pressure inside the PDMS chamber reaching 28 kPa with piston activation. COMSOL file used to model MitoPunch pressure is available in *Figure 1—source data 1*. (**C**) Schematic of MitoCeption technique. Recipient cells ($1 \times 10^5$) are seeded on wells of a 6-well dish ~24 hr before delivery. A freshly isolated suspension of mitochondria in 1× DPBS with calcium and magnesium, pH 7.4, is pipetted into the cell medium before the plate is centrifuged at $1500 \times g$ for 15 min at 4°C. The plate is incubated in a 37°C incubator for 2 hr before being centrifuged again at $1500 \times g$ for 15 min at 4°C. (**D**) MitoCeption pressure model and calculated pressure exerted by isolated mitochondria on recipient cells during delivery.
The online version of this article includes the following source data and figure supplement(s) for figure 1:

**Source data 1.** Numerical simulation of MitoPunch pressure generation during mitochondrial delivery.
**Figure supplement 1.** Annotated MitoPunch apparatus.

internalization of delivered mitochondria. These data suggest that the method of delivery and target cell type affect the efficiency of initiating mitochondria–recipient cell interactions.

We quantified the number of discreet dsRed-spots in each cell ~2 hr following delivery from this data (*George et al., 2004*; *Figure 2B* and *Figure 2—figure supplement 1*). ImageStream spot count analysis of 143BTK– ρ0 recipient cells showed MitoPunch delivered a lower mean and median number of dsRed spots per cell than coincubation or MitoCeption. MitoPunch transfers into BJ ρ0 recipient cells yielded fewer mean spots/cell compared to coincubation and MitoCeption with an equivalent median number of spots/cell for MitoPunch and MitoCeption and slightly more for coincubation. Next, we used confocal microscopy to observe dsRed mitochondrial fluorescence in 143BTK– ρ0 recipients fixed 15 min post-transfer, which we chose for its robust mitochondrial acquisition (*Figure 2C*). We visualized mitochondrial localization with confocal microscopy by detecting dsRed protein from the donor mitochondria, shown in red, and labeling the recipient cell plasma membranes with either CellMask Green (coincubation and MitoCeption) or wheat germ agglutinin (MitoPunch), shown in green. Following MitoPunch, mitochondrial dsRed appeared to localize to pores in the filter insert and within the cytoplasm of cells, whereas coincubation and MitoCeption uniformly coated recipient cells with mitochondria, with greater mitochondrial association with recipient cells following MitoCeption. While all three methods initiate physical interactions between mitochondria and recipient cells, MitoPunch delivers mitochondria to the basal membranes of recipient cells at regions associated with the PET membrane pores, compared to a diffuse membrane association pattern seen with coincubation and MitoCeption.

To investigate the capacity of these methods to disrupt recipient cell plasma membranes, we delivered the membrane impermeant dye propidium iodide (PI) by coincubation, MitoPunch, and MitoCeption to measure membrane disruption from delivery and quantified uptake by flow cytometry (*Figure 2D*; *Novickij et al., 2017*). Delivery into 143BTK– ρ0 cells by MitoPunch and MitoCeption

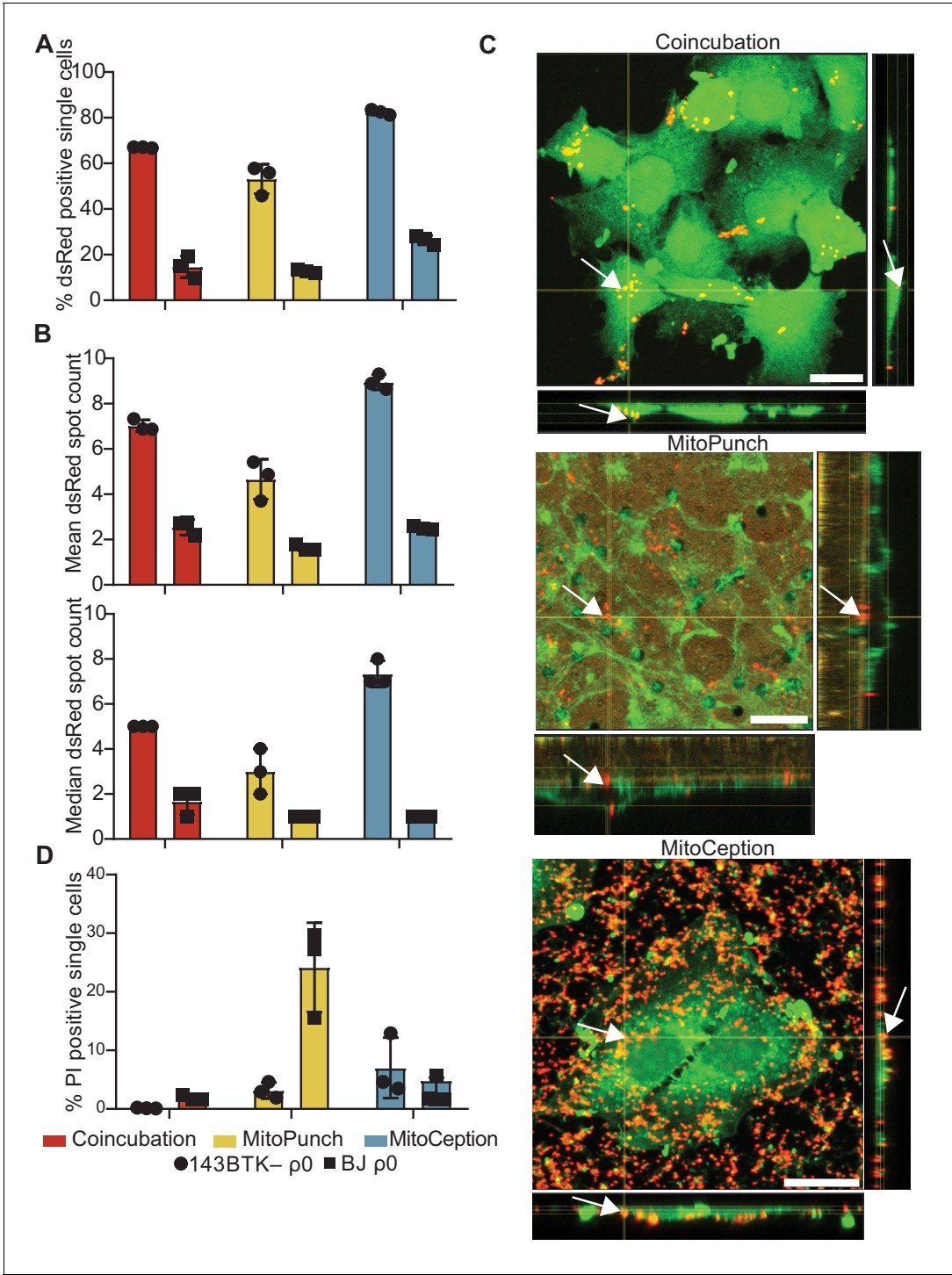

**Figure 2.** MitoPunch delivers isolated mitochondria to recipient cells. (**A**) Quantification of flow cytometry results measuring the association of dsRed mitochondria with 143BTK– ρ0 and BJ ρ0 single recipient cells following mitochondrial transfer. (**B**) Mean and median dsRed spot count quantification of ImageStream data. (**C**) Sequential Z-stacks of confocal microscopy of 143BTK– ρ0 cells delivered isolated HEK293T-derived dsRed mitochondria by coincubation, MitoPunch, and MitoCeption and fixed 15 min following transfer. Arrows indicate representative mitochondria interacting with recipient cells. Transferred dsRed mitochondria are labeled in red. Plasma membranes are labeled in green, stained with CellMask Green plasma membrane stain in coincubation and MitoCeption and with wheat germ agglutinin plasma membrane stain in MitoPunch. Scale bars indicate 15 μm. (**D**) Quantification of flow cytometry measurements of fluorescence in 143BTK– ρ0 and BJ ρ0 single cells following propidium iodide transfer by coincubation, MitoPunch, and MitoCeption. Error bars represent SD of three technical replicates in all figures.

The online version of this article includes the following figure supplement(s) for figure 2:

*Figure 2 continued on next page*

*Figure 2 continued*

**Figure supplement 1.** Mitochondrial spot quantification.

resulted in similar percentages of PI-positive recipient cells, and both were greater than coincubation. Interestingly, BJ ρ0 cells showed comparable fractions of PI-positive cells to the 143BTK– ρ0 after coincubation and MitoCeption. However, MitoPunch yielded an approximately fivefold increase in the PI-positive fraction compared to all other conditions. These data show that MitoPunch and MitoCeption disrupt the plasma membranes of recipient cells for potential mitochondrial transfer, and the degree of disruption is cell type and delivery method dependent.

## Stable retention of transplanted mtDNA

After verifying mitochondrial interaction with recipient cells by coincubation, MitoPunch, and MitoCeption, we next determined whether these methods result in permanent retention of exogenous mtDNA to generate SIMR cells. ρ0 cells cannot synthesize pyrimidines and therefore cannot proliferate or survive without supplemented uridine because of ETC impairment, so we used nucleotide-free medium prepared with dialyzed fetal bovine serum (SIMR selection medium) to select for SIMR cells with transplanted mtDNA and restored ETC activity (*Grégoire et al., 1984*; *Figure 3A and B*).

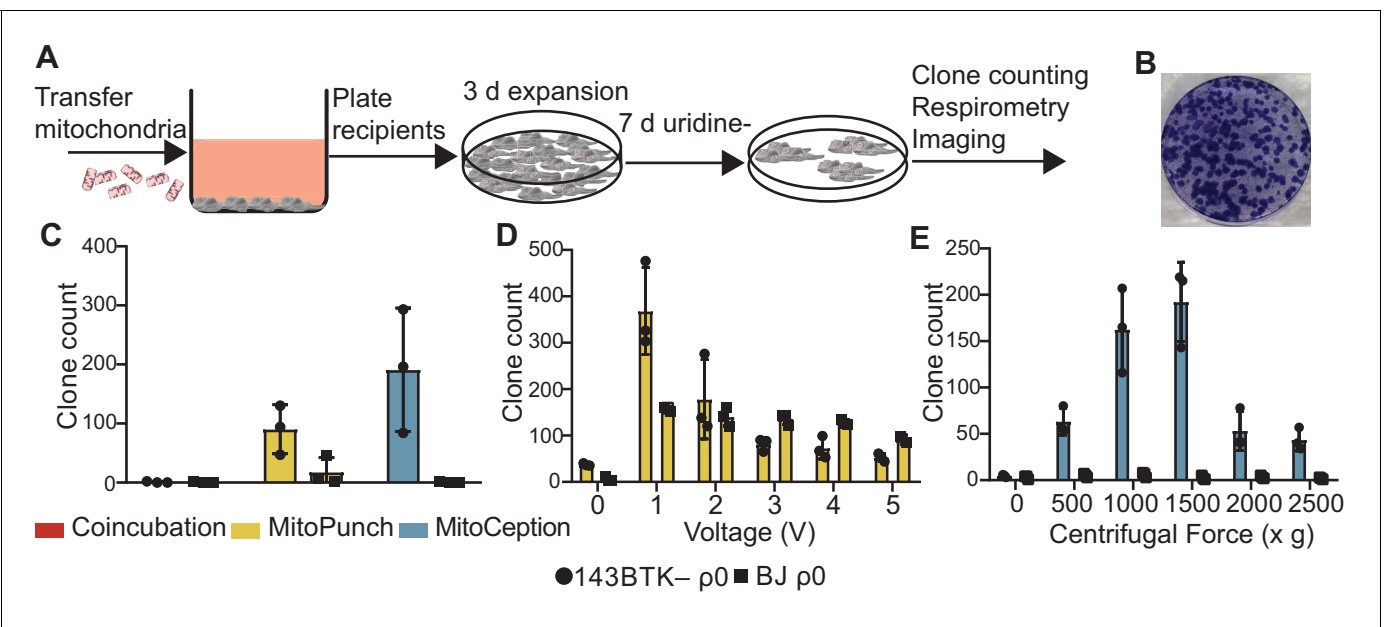

**Figure 3.** Stable retention of transplanted mitochondrial DNA (mtDNA) into transformed and replication-limited cells. (**A**) Workflow for stable isolated mitochondrial recipient (SIMR) cell generation by mitochondrial transfer into ρ0 cells. (**B**) Representative fixed and crystal violet stained 10 cm plate image following MitoPunch and SIMR cell selection used for SIMR clone generation quantification. (**C**) Quantification of crystal violet stained 143BTK– ρ0 and BJ ρ0 SIMR clones. Error bars represent SD of three technical replicates. (**D**) Quantification of crystal violet stained 143BTK– ρ0 and BJ ρ0 SIMR clones formed by MitoPunch actuated with indicated voltages after uridine-free selection. Error bars represent SD of three technical replicates with the exception of BJ ρ0 5 V transfer, which shows two replicates. (**E**) Quantification of crystal violet stained 143BTK– ρ0 and BJ ρ0 SIMR clones formed by MitoCeption with indicated centripetal forces after uridine-free selection. Error bars represent SD of three technical replicates.

The online version of this article includes the following figure supplement(s) for figure 3:

**Figure supplement 1.** Verification of surviving mitochondrial donor cells following mitochondrial isolation.

**Figure supplement 2.** MitoPunch generates stable isolated mitochondrial recipient (SIMR) clones in immortalized mouse cells.

**Figure supplement 3.** Quantification of MitoPunch reproducibility.

**Figure supplement 4.** Quantification of MitoPunch reproducibility relative to mitochondrial mass transferred.

**Figure supplement 5.** Quantification of stable isolated mitochondrial recipient (SIMR) generation efficiency by delivering different masses of isolated mitochondria.

**Figure supplement 6.** Quantification of MitoPunch stable isolated mitochondrial recipient (SIMR) generation by serial deliveries using one isolated mitochondrial aliquot.

BJ ρ0 cells survive longer under this selection scheme compared to the 143BTK– ρ0 (data not shown), so we included an additional selection phase by culturing these cells in nucleotide-free, glucose-free, galactose supplemented medium (galactose selection medium) (*Robinson et al., 1992*). We isolated and transferred HEK293T dsRed mitochondria into 143BTK– ρ0 and BJ ρ0 cells by coincubation, MitoPunch, and MitoCeption, performed SIMR selection in cell-type appropriate medium for 7 days, and quantified the number of viable clones by crystal violet staining (*Figure 3C*). Coincubation did not generate SIMR clones in 143BTK– ρ0 cells, in contrast to MitoPunch and MitoCeption, which each generated dozens of clones. BJ ρ0 cells with delivered HEK293T mitochondria by coincubation or MitoCeption did not form SIMR clones. MitoPunch generated numerous SIMR clones in both cell types, although fewer BJ ρ0 SIMR clones than in 143BTK– ρ0 cells, whereas MitoCeption only generated clones in 143BTK– ρ0 cells and was unable to form stable clones in replication-limited BJ cells. To assess the risk of mitochondrial donor cells surviving disruption during mitochondrial isolation and generating false positive SIMR clones, we performed three independent mitochondrial isolations, plated an aliquot from each isolation representing mitochondria isolated from ~$1.5 \times 10^7$ HEK293T dsRed cells on 10 cm dishes, and carried these plates through the 10-day selection with SIMR selection medium before crystal violet staining for visual assessment (*Figure 3—figure supplement 1*). We observed no cell growth on any of the three plates, indicating a minimal incidence of donor cell survival through the mitochondrial isolation protocol.

We next investigated whether differences in SIMR clone generation between 143BTK– ρ0 and BJ ρ0 cells were driven by sensitivity to differences in delivery pressure. We developed a MitoPunch device with adjustable plunger acceleration modulated by changing the circuit voltage (ImmunityBio). We generated independent voltage titration curves in 1 V increments (0 V – 5 V) for each cell type in technical triplicate at each voltage and used the same mitochondrial preparation for all samples for each cell type. All prior experiments in this study are controlled by delivering DPBS with calcium and magnesium to recipient cells by MitoPunch, but here we included a 0 V condition in which the seeded filter insert was positioned atop the PDMS reservoir and pressed against an aliquot of isolated mitochondrial suspension similar to deliveries with force, but without actuating the piston. We achieved maximum 143BTK– ρ0 SIMR clone generation with this tunable MitoPunch at 1 V, with a sharp reduction to background with increasing voltage (*Figure 3D*). The BJ ρ0 recipient also showed maximal SIMR generation at 1 V, with a shallow decline in SIMR generation efficiency to 5 V. Surprisingly, the 0 V condition consistently yielded a few SIMR clones in the 143 BTK– ρ0 recipients and inconsistently in the BJ ρ0 recipients. This result suggests that the pressure generated by sealing the filter insert against the PDMS reservoir is sufficient to generate SIMR clones at a low frequency. For all forthcoming MitoPunch trials we use the variable voltage MitoPunch device set to 1 V.

We performed a similar force titration with MitoCeption by varying the maximum centripetal force, using a common mitochondrial preparation for all samples of both cell types. In 143BTK– ρ0 cells, we observed maximum clone generation at $1000 \times g$ and $1500 \times g$, and we did not generate BJ ρ0 SIMR clones greater than the $0 \times g$ background at any acceleration tested (*Figure 3E*). This background, present in both 143BTK– ρ0 and BJ ρ0 conditions at $0 \times g$, is likely from rare un-lysed donor cells from mitochondrial preparations directly pipetted into the culture medium of recipient cells during MitoCeption. We have infrequently observed imperfect donor cell lysis , usually in larger mitochondrial preparations, that results in rare, persistant dsRed fluorescent colonies as observed by fluorescence microscopy. True SIMR clones cannot produce the dsRed protein from donor mitochondria and lose fluorescence with time over selection, while these persistent dsRed colonies maintain their fluorescence over the same period (data not shown). Despite this occasional low-level contaminating donor cell background, MitoCeption yielded a strong dose-dependent response in SIMR clone generation from 143BTK– ρ0 recipients above the background. Additionally, MitoPunch deliveries into B16 ρ0 mouse melanoma cells (*Tan et al., 2015*) yielded maximal SIMR generation at a different voltage than in the human cell lines tested, showing that optimal mitochondrial delivery pressure may be cell type dependent (*Figure 3—figure supplement 2*). These data suggest that MitoPunch is uniquely able to generate SIMR clones in replication-limited fibroblasts and SIMR generation efficiency depends on delivery pressure.

We next quantified the reproducibility of our mitochondrial preparation technique and the MitoPunch procedure by performing triplicate MitoPunch transfers using three independent mitochondrial preparations from equal numbers of HEK293T dsRed biological replicate populations (*Figure 3—figure supplements 3* and *4*). We define biological replication here as mitochondrial

preparations derived from independently cultured populations of mitochondrial donor cells. Mitochondrial preparations 1, 2, and 3 (same as those pictured in *Figure 3—figure supplement 1*) generated consistent protein concentrations, and each preparation yielded dozens of SIMR clones in all three technical replicate MitoPunch deliveries with the exception of Prep 3, which resulted in two lower efficiency replicates. We quantified the number of SIMR clones generated per microgram of mitochondrial mass loaded into the MitoPunch apparatus and observed a similar trend. These results showed that our mitochondrial isolation technique produced consistent levels of isolated mitochondrial mass and that the MitoPunch technique yielded high numbers of SIMR clones.

To enable desirable mtDNA–nDNA clone generation using limited starting material, such as mitochondria from rare cell subpopulations, we determined the minimal mass of mitochondrial isolate required to generate SIMR clones. We performed coincubation, MitoPunch, and MitoCeption transfers into ~1 × 10$^5$ 143BTK– ρ0 recipient cells using decreasing concentrations of dsRed mitochondria isolated from HEK293T cells and plated half of the recipient cell population on 10 cm plates. We observed a similar dose-dependent relationship between mitochondrial mass delivered and SIMR clones observed for MitoPunch and MitoCeption across 0.16 μg, 1.6 μg, and 16 μg total mitochondrial protein suspended in 120 μL of 1× DPBS, pH 7.4 transfer buffer (*Figure 3—figure supplement 5*). These results showed that although MitoPunch and MitoCeption generate SIMR clones from transformed recipient cells with similar efficiency per microgram of mitochondrial isolate delivered, the differences inherent to the two protocols rendered direct comparisons of their relative efficiencies less meaningful.

Moving the seeded PET filter from a 12-well dish to the MitoPunch apparatus often resulted in excess medium being carried to the PDMS reservoir. Combined with the small volume of mitochondrial preparation delivered to the recipient cells, we observed that MitoPunch resulted in diluted residual mitochondrial isolate left in the reservoir post-transfer. In the interest of conserving mitochondrial material, we tested whether a used 120 μL aliquot of isolated mitochondria can be applied to repeated MitoPunch transfers to generate SIMR clones (*Figure 3—figure supplement 6*). We performed 11 sequential deliveries into 143BTK– ρ0 cells using one aliquot of mitochondrial isolate and found maximal SIMR clone generation from the first and second deliveries, after which we observe a sharp reduction in SIMR cell formation and inconsistent SIMR generation rate up to the 11th transfer. These data showed that multiple MitoPunch transfers can be performed using a single aliquot of mitochondrial suspension when material is limited.

## SIMR cells rescue ρ0 mitochondrial respiration and network morphology

Finally, we measured mitochondrial function in SIMR cells by quantifying the rate of oxygen consumption and assessing mitochondrial morphology. We isolated three independent 143BTK– ρ0 SIMR clones generated by MitoPunch or MitoCeption transfer of isolated HEK293T mitochondria and measured each clone's oxygen consumption rate (OCR) using a Seahorse Extracellular Flux Analyzer mitochondrial stress test (*Figure 4A*, *Figure 4—figure supplement 1*). To determine whether SIMR clone respiration remained stable through time, we grew the clones through two freeze/thaw cycles in uridine supplemented medium and measured cellular respiration. We found that one MitoCeption clone lost its respiratory capacity and one MitoPunch clone was not viable after freezing and thawing (data not shown). In the remaining clones, basal and maximal respiration, spare respiratory capacity, and ATP generation remained stable throughout both freeze-thaw cycles. We have performed numerous similar experiments using a range of recipients and mitochondrial donors and observed successful clone viability after freeze-thaw (data not shown).

We then immunostained the freeze-thawed SIMR clones with anti-TOM20 and anti-double-stranded DNA (dsDNA) antibodies to detect mitochondria and mtDNA content, respectively, by confocal microscopy (*Figure 4B* and *Figure 4—figure supplement 2*). The MitoCeption clone that lost respiratory capacity showed a fragmented mitochondrial network with no detectable mtDNA (*Figure 4—figure supplement 2*), whereas the other SIMR clones generated by MitoPunch and MitoCeption contained mtDNA with filamentous mitochondrial network morphologies. These data show that the majority of 143BTK– ρ0 SIMR clones generated by either MitoPunch or MitoCeption have retained mtDNA, restored respiratory profiles, and filamentous mitochondrial network morphologies.

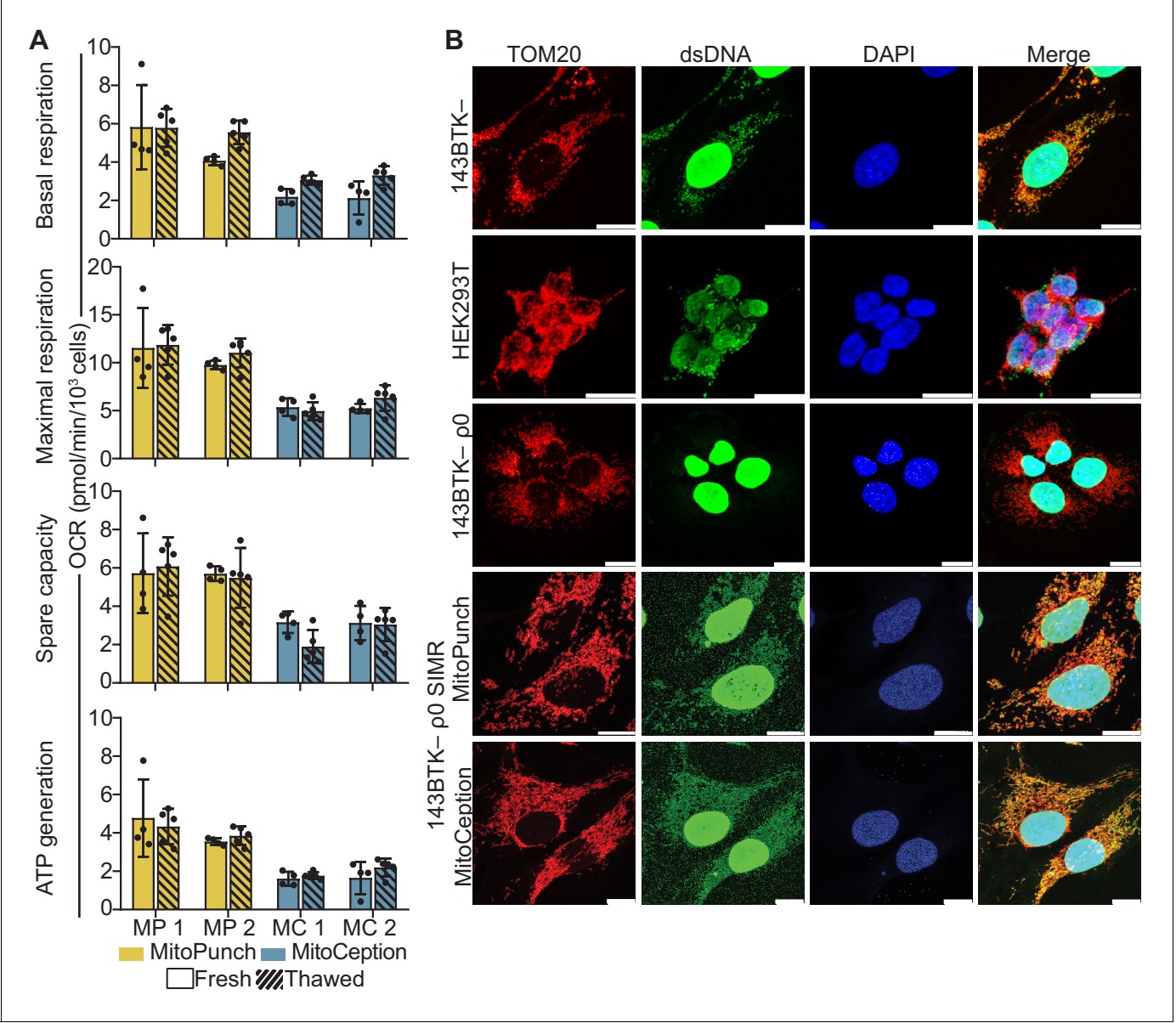

**Figure 4.** Mitochondrial DNA (mtDNA) transplantation rescues ρ0 mitochondrial phenotypes. (**A**) Oxygen consumption rate (OCR) quantification of basal and maximal respiration, spare respiratory capacity, and ATP generation from two independent 143BTK– ρ0 + HEK293T stable isolated mitochondrial recipient (SIMR) clones generated by MitoPunch and MitoCeption. Cross-hatched data indicate clones that were frozen and thawed twice each. Error bars represent SD of four technical replicates for fresh SIMR cell measurements and five for thawed SIMR cell measurements. (**B**) Confocal microscopy of representative 143BTK– ρ0 + HEK293T SIMR clones compared to 143BTK– parental, HEK293T dsRed mitochondrial donor, and 143BTK– ρ0 controls. Mitochondria were stained with anti-TOM20 antibody and labeled red, double-stranded DNA was stained with anti-dsDNA antibody and labeled green, and cell nuclei were stained with NucBlue (Hoechst 33342) and labeled blue. Scale bars indicate 15 μm.

The online version of this article includes the following figure supplement(s) for figure 4:

**Figure supplement 1.** Schematic of the Seahorse Mito Stress Test.

**Figure supplement 2.** Confocal microscopy of stable isolated mitochondrial recipient (SIMR) lines.

## Discussion

Stability of the mitochondrial genome is essential for studying the long-term effects of mtDNA–nDNA interactions and for potential therapeutic applications of mitochondrial transfer. MitoPunch generates up to hundreds of SIMR clones in both transformed and Hayflick-limited recipient cells by exerting a pressure sufficient to perforate mammalian cell membranes in regions small enough to be

repaired within minutes, which sustains cell viability and resumed cell growth and proliferation (*Boye et al., 2017*). We have generated SIMR clones by MitoPunch with mitochondria isolated by a commercially available kit, as performed here, as well as by using standard mitochondrial isolation buffers. Additionally, we achieved similar results by disrupting mitochondrial donor cells using Dounce homogenization (data not shown) but found the commercially available kit with syringe disruption is advantageous due to its ease of use, reproducibility, and a reduced number of steps to isolate mitochondria. We generated dozens of SIMR clones by MitoPunch and MitoCeption using these mitochondrial isolation methods and anticipate that other mitochondrial preparation techniques will also yield SIMR clones.

Interestingly, we do not observe SIMR clone generation by coincubation in our study. Few reports show limited stable clone formation by coincubation techniques, but these studies used up to 100-fold higher levels of exogenous mitochondria in coincubation experiments than required for Mito-Punch or MitoCeption in our hands, or antibiotic selection schemes to achieve stable mitochondrial transfer (*Clark and Shay, 1982*; *Patel et al., 2017*). High levels of mitochondrial protein are easily isolated from fast-growing immortalized cell lines but may not be available when using human donor-derived or other limiting starting material. Additionally, mitochondrial donor cells of interest nearly exclusively lack antibiotic selection markers, making such selection schemes unfeasible. Particularly in those cases, the greatly enhanced SIMR generation capacity of MitoPunch and MitoCeption is strongly enabling for generating desired mtDNA–nDNA combinations.

The distinct mechanisms and procedures of MitoPunch and MitoCeption make direct comparisons of their relative efficiencies challenging. Despite this, our results demonstrate that both techniques generate SIMR clones from ρ0 transformed cells in a mitochondrial dose-dependent fashion and can be readily adopted by laboratories studying mtDNA–nDNA interactions. Strikingly, in the cell types we have tested, we find that only MitoPunch generates SIMR clones from ρ0 primary, non-immortal cells. Studies in our laboratory suggest that the transcriptome and metabolome of replication-limited SIMR clones differ significantly from un-manipulated control clones but can be recovered and reset to un-manipulated control levels by cellular reprogramming to induced pluripotent stem cells and subsequent differentiation (*Patananan et al., 2020*). These results indicate that SIMR clone generation in replication-limited, reprogrammable cells is crucial for studies of mtDNA–nDNA interactions involving mitochondrial transplantation into ρ0 cells, and that MitoPunch is uniquely capable of efficiently generating enough clones for statistically valid studies in such work. We have circumvented the need for ρ0 recipient cells by using the MitoPunch technology to completely replace mutant mtDNA in mouse cells without mtDNA depletion. This was done by delivering mitochondria containing mtDNA with a chloramphenicol resistant point mutation and selecting for SIMR clones containing only rescue mtDNA using antibiotic supplemented nucleotide-free medium (*Dawson et al., 2020*). However, this workflow is dependent upon using antibiotic resistant mitochondrial donor cells and is not applicable to investigating the full spectrum of mtDNA sequences required for robust studies of mtDNA–nDNA interactions. Future work with MitoPunch and other isolated mitochondrial transfer modalities will be improved by developing techniques to avoid fully depleting the mtDNA of recipient cells of interest before generating SIMR clones for downstream analysis and applications.

# Materials and methods

**Key resources table**

| Reagent type (species) or resource | Designation | Source or reference | Identifiers | Additional information |
|---|---|---|---|---|
| Cell line (*Homo sapiens*) | 143 BTK– ρ0 osteosarcoma | *Patananan et al., 2020*, ATCC | Cat. #CRL-8303; RRID:CVCL_9W36 | |
| Cell line (*Homo sapiens*) | 143 BTK– osteosarcoma | ATCC | Cat. #CRL-8303; RRID:CVCL_9W36 | |
| Cell line (*Homo sapiens*) | BJ ρ0 foreskin fibroblast (male) | *Patananan et al., 2020* ATCC | Cat. #CRL-2522; RRID:CVCL_3653 | |

*Continued on next page*

*Continued*

| Reagent type (species) or resource | Designation | Source or reference | Identifiers | Additional information |
|---|---|---|---|---|
| Cell line (*Homo sapiens*) | HEK293T dsRed | *Miyata et al., 2014* | | A gift from the laboratory of Dr. Carla Koehler |
| Cell line (*M. musculus*) | B16 ρ0 melanoma | *Dong et al., 2017* | | A gift from the laboratory of Dr. Michael Berridge |
| Cell line (*M. musculus*) | L929 fibroblasts | ATCC | Cat. #CCLl-1 | |
| Antibody | Anti-TOMM20 (Rabbit monoclonal) | Abcam | Cat. #ab78547 RRID:AB_2043078 | IF(1:1000) |
| Antibody | Anti-dsDNA (Mouse monoclonal) | Abcam | Cat. #ab27156 RRID:AB_470907 | IF(1:1000) |
| Antibody | Anti-rabbit IgG (Donkey polyclonal) | Thermo Fisher Scientific | Cat. #A-31573 RRID:AB_2536183 | IF(1:100) |
| Antibody | Anti-mouse IgG (Donkey polyclonal) | Thermo Fisher Scientific | Cat. #A-21202 RRID:AB_141607 | IF(1:100) |
| Commercial assay or kit | Qproteome Mitochondria Isolation kit | Qiagen | Cat. #37612 | |
| Commercial assay or kit | BCA protein assay | Thermo Fisher | Cat. #23225 | |
| Chemical compound, drug | Propidium iodide | Thermo Fisher Scientific | Cat. #P1304MP | |
| Chemical compound, drug | Accutase | Thermo Fisher Scientific | Cat. #A1110501 | |
| Chemical compound, drug | 16% paraformaldehyde | Thermo Fisher Scientific | Cat. #28906 | |
| Chemical compound, drug | Triton-X 100 | Sigma | Cat. #X100 | |
| Chemical compound, drug | ProLong Gold Antifade Mountant with DAPI | Invitrogen | Cat. #P3691 | |
| Chemical compound, drug | ProLong Glass Antifade Mountant with NucBlue Stain | Thermo Fisher Scientific | Cat. #P36985 | |
| Chemical compound, drug | Uridine | Thermo Fisher Scientific | Cat. #AC140770250 | |
| Chemical compound, drug | Galactose | Sigma-Aldrich | Cat. #G5388-100G | |
| Chemical compound, drug | CellMask Green PM | Molecular Probes | Cat. #C37608 | |
| Chemical compound, drug | Alexa Fluor 488 conjugated Wheat Germ Agglutinin | Invitrogen | Cat. #W11261 | |
| Chemical compound, drug | Crystal violet | Thermo Fisher Scientific | Cat. #C581-25 | |
| Software, algorithm | Wave 2.6.2 | Agilent | RRID:SCR_014526 | |
| Software, algorithm | FlowJo 10.6.2 | BD Biosciences | RRID:SCR_008520 | |
| Software, algorithm | IDEAS 6.2 | Luminex | | |
| Software, algorithm | Multiphysics 5.3 | COMSOL | RRID:SCR_014767 | |
| Software, algorithm | Imaris Viewer 9.5.1 | Oxford Instruments | RRID:SCR_007370 | |
| Software, algorithm | Imaris File Converter 9.5.1 | Oxford Instruments | RRID:SCR_007370 | |
| Software, algorithm | Prism v.8 | Graphpad | RRID:SCR_002798 | |
| Software, algorithm | LAS X Lite 3.7.1.21655 | Leica | | |
| Software, algorithm | FIJI | *Schindelin et al., 2012* | | |
| Other | Dialyzed FBS | Life Technologies | Cat#26400–044 | |

*Continued*

| Reagent type (species) or resource | Designation | Source or reference | Identifiers | Additional information |
|---|---|---|---|---|
| Other | 12–well 3.0 μm Transparent PET Membrane | Corning | Cat#353181 | |
| Other | Glass coverslips | Zeiss | Cat#474030–9000 | |
| Other | V3 96-well plate | Agilent | Cat#101085–004 | |
| Other | Variable voltage MitoPunch apparatus | ImmunityBio | | Inquiries regarding this device can be made to the corresponding author |

## Cell culture conditions

Human ρ0 cells were grown in DMEM (Fisher Scientific, Waltham, MA, Cat. # MT10013CM) supplemented with 10% FBS, non-essential amino acids (Gibco, Waltham, MA, Cat. #11140–050), GlutaMax (Thermo Fisher Scientific, Waltham, MA, Cat. # 35050–061), penicillin and streptomycin (VWR, Radnor, PA, Cat. # 45000–652), and 50 mg/L uridine (Thermo Fisher Scientific, Cat. # AC140770250). All other human cell lines were grown in DMEM (Fisher Scientific, Cat. # MT10013CM) supplemented with 10% FBS, non-essential amino acids, GlutaMax, and penicillin and streptomycin. B16 ρ0 cells were grown in RPMI (Thermo Fisher Scientific, Cat. # MT-10–040 CM) supplemented with 10% FBS, non-essential amino acids, GlutaMax, penicillin and streptomycin, pyruvate (Corning, Corning, NY, Cat. # 25000 CI), and 50 mg/L uridine. L929 cells were grown in RPMI supplemented with 10% FBS, non-essential amino acids, GlutaMax, penicillin and streptomycin, and pyruvate. All mammalian cells were cultured in a humidified incubator maintained at 37°C and 5% $CO_2$. The following cells were used in this study: HEK293T dsRed (female), 143BTK– (female), 143BTK– ρ0 (female), BJ ρ0 (male), B16 (male), and L929 (male). We have not formally identified these cell lines; however, we have sequenced their mitochondrial and nuclear DNA for polymorphisms and find unique sequences which we use for genotyping our cultures (unpublished data). BJ ρ0 cells were used as mitochondrial recipients within three passages of thaw for all mitochondrial transfer experiments in this work to avoid the onset of senescence. All lines were routinely tested for mycoplasma with negative results.

## Mitochondrial isolation

Mitochondria were isolated from ~$1.5 \times 10^7$ mitochondrial donor cells per mitochondrial transfer using the Qproteome Mitochondrial Isolation Kit (Qiagen, Hilden, Germany, Cat. #37612) with slight alterations to the manufacturers protocol. Mitochondrial donor cells were harvested using a cell scraper (Fisher Scientific, Cat. # 08-100-241) and collected in 50 mL conical tubes at approximately $6 \times 10^7$ cells per tube (Thermo Scientific, Cat. #12-565-271). Cells were pelleted by centrifugation at $500 \times g$ for 10 min at 4°C and washed with DPBS before pelleting again by centrifugation at $500 \times g$ for 10 min at 4°C. Cells were resuspended at $1 \times 10^7$ cells/mL in ice-cold Lysis Buffer with Protease Inhibitor Solution and incubated for 10 min at 4°C in 2 mL tubes on an end-over-end shaker. Lysates were centrifuged at $1000 \times g$ for 10 min at 4°C and supernatant was aspirated. Pellets were resuspended in 1.5 mL ice-cold Disruption Buffer with Protease Inhibitor Solution and mechanical disruption was accomplished by 10 passes through a 26 G blunt ended needle (VWR, Radnor, PA, Cat. # 89134–164) attached to a 3 mL syringe (VWR, Cat. # BD309657). The subsequent lysates were centrifuged at $1000 \times g$ for 10 min at 4°C and the supernatants were transferred to new 2 mL tubes. The resultant supernatants were centrifuged again at $1000 \times g$ for 10 min at 4°C to remove any remaining intact cells, and the supernatants were transferred to clean 1.5 mL tubes. These supernatants were centrifuged at $6000 \times g$ for 10 min at 4°C and the supernatants were aspirated. The resulting mitochondrial pellets were resuspended in mitochondrial storage buffer and pelleted by centrifugation at $6000 \times g$ for 20 min at 4°C. The isolated mitochondrial pellets were resuspended in 120 μL per transfer replicate 1× DPBS with calcium and magnesium (Thermo Fisher Scientific, Cat. # 14040133) immediately prior to mitochondrial transfer and kept on ice.

## Mitochondrial coincubation

~1 × 10$^5$ 143BTK– ρ0 or BJ ρ0 cells were seeded into wells of 6-well dishes ~ 24 hr prior to delivery. Mitochondria isolated from ~1.5 × 10$^7$ HEK293T dsRed cells resuspended in 120 μL 1× DPBS with calcium and magnesium were pipetted into the culture medium of each well containing recipient cells and incubated at 37°C and 5% $CO_2$ for 2 hr. Cells were then released from the dish using Accutase (Thermo Fisher Scientific, Cat. # A1110501) and seeded into 10 cm plates for SIMR cell selection or harvested for additional analyses.

## MitoPunch apparatus construction

A 5 V solenoid (Sparkfun, Boulder, CO, Cat. # ROB-11015) is screwed into a threaded plug (Thor Labs, Newton, NJ, Cat. # SM1PL) and inserted into a bottom plate (Thor Labs, Cat. # CP02T) (*Figure 1—figure supplement 1*). The solenoid is regulated by a Futurlec mini board (Futurlec, New York, NY, Cat. # MINIPOWER) and powered by a MEAN WELL power supply (MEAN WELL, New Taipei City, Taiwan, Cat. # RS-35–12). Optomechanical assembly rods (Thor Labs, Cat. # ER3) are inserted into the bottom plate. The middle and top plates (Thor Labs, Cat. # CP02) are threaded through the assembly rods. The middle plate is fitted with a retaining ring, which supports an aluminum washer (outer diameter, 25 mm; inner diameter, 10 mm). The middle plate is secured along the assembly rods using the included screws. The retaining ring is adjusted such that the top surface of the washer is at the same height as the piston surface in its retracted state. A flexible PDMS (10:1 ratio of Part A base: Part B curing agent) (Fisher Scientific, Cat. #NC9644388) reservoir consisting of a bottom layer (25 mm diameter, 0.67 mm height) bonded to an upper ring (outer diameter, 25 mm; inner diameter, 10 mm; height, 1.30 mm) is placed on top of the washer. This reservoir can contain up to ~120 μL of liquid. To perform MitoPunch delivery, a 3 μm membrane transwell insert (Corning, Cat. # 353181) seeded with 1 × 10$^5$ adherent cells is lowered through the top plate and rested atop one retaining ring. The insert is secured to the top plate by an additional retaining ring. This assembly is lowered until the base of the insert contacts the top surface of the PDMS reservoir and is secured in place with screws to form a tight seal. In addition, a variable voltage version of this device based on the same principles with identical delivery procedures as MitoPunch, but with tunable plunger acceleration achieved by varying actuator voltage, was engineered by ImmunityBio and is available upon request to the corresponding author. Optimal MitoPunch delivery voltage for individual cell lines is determined empirically by performing a voltage-response curve in technical triplicate across a range of voltages from 1 V to 5 V using the piston acceleration control software.

## Seeding cells for MitoPunch mitochondrial transfer

Filter inserts with 3 μm pores (Corning, Cat. # 353181) are placed in wells of a 12-well dish. 1.5 mL warm uridine supplemented medium is dispensed in the wells outside of the filter insert, and 1 × 10$^5$ adherent cells suspended in 0.5 mL warm uridine supplemented medium are seeded within the filter inserts and placed in a humidified incubator maintained at 37°C and 5% $CO_2$ 1 day prior to mitochondrial delivery.

## MitoPunch mitochondrial transfer

Following mitochondrial isolation, the MitoPunch apparatus is sterilized with 70% ethanol and entered into the biological safety cabinet and an autoclaved PDMS reservoir is placed in the device as indicated in *Figure 1—figure supplement 1*. The PDMS reservoir is washed 3× with 120 μL sterile DPBS with calcium and magnesium after being set in the MitoPunch apparatus. 120 μL mitochondrial suspension from ~1 × 10$^7$ donor cells in DPBS with calcium and magnesium is loaded into the PDMS reservoir. Mitochondrial transfer is performed by securing the seeded membrane to the PDMS reservoir and actuating the solenoid for 3 s. The mechanical plunger strikes the middle of the PDMS chamber, displacing the base layer by ~1.3 mm. This displacement pressurizes the mitochondrial suspension and propels it through the membrane and into the cells (*Figure 1B*). Once the solenoid has returned to its starting position, the insert is removed from the apparatus, placed back in the 12-well dish in its original medium, and incubated at 37°C and 5% $CO_2$ for 2 hr. Cells were then released from the dish using Accutase and seeded into 10 cm plates for SIMR cell selection or harvested for additional analyses.

### Collecting mitochondrial recipient cells following MitoPunch transfer

Following MitoPunch mitochondrial transfer and 2 hr incubation, medium is aspirated from within the transwell filter with care taken not to disrupt the cells on the membrane, and then from outside and underneath the filter insert. The well and insert are washed 1× with DPBS (0.5 mL inside the insert and 1 mL outside the insert) with DPBS aspirated as before. Cells are released from the membrane by 5 min incubation at 37°C and 5% $CO_2$ with Accutase (0.5 mL inside the insert and 1 mL outside the insert). Following incubation, the cells are suspended in the Accutase within the filter insert using a P1000 pipette, being careful not to puncture the PET membrane, and directly pipetted into 10 cm plates with 10 mL warm uridine supplemented medium.

### MitoCeption

As described previously (*Caicedo et al., 2015*), $1 \times 10^5$ recipient cells were seeded in each well of a 6-well dish and incubated at 37°C and 5% $CO_2$ overnight. Mitochondrial isolate from ~$1 \times 10^7$ donor cells suspended in 1× DPBS with calcium and magnesium was pipetted into the well and the plate was centrifuged at 1500 × *g* for 15 min at 4°C. Cells were removed from the centrifuge and incubated for 2 hr at 37°C and 5% $CO_2$ before being centrifuged a second time at 1500 × *g* for 15 min at 4°C. Cells were then released from the dish using Accutase and seeded into 10 cm plates for SIMR cell selection or harvested for additional analyses.

The pressure generated by the MitoCeption method was estimated by calculating the force exerted per unit area of the cell membrane during centrifugation. The force induced by the centrifugation of a single mitochondrion on the cell membrane was equal to the centripetal force of the mitochondria under the acceleration of 1500 × *g* minus the buoyancy force,

$$F_{centrifugation} = (m_{mito} - m_{water}) * a$$

where $m_{mito}$ and $m_{water}$ are the mass of mitochondria and water, and $a$ is the acceleration rate of centrifugation. The equivalent pressure induced by mitochondria during centrifugation was approximated by

$$p = \frac{F_{centrifugation}}{S} = \frac{(m_{mito} - m_{water}) * a}{S} = \frac{(\rho_{mito} - \rho_{water})Va}{S} \approx (\rho_{mito} - \rho_{water}) * a * d$$

where $\rho_{mito}$ (1.1 g/cm³) and $\rho_{water}$ (1.0 g/cm³) are the density of mitochondria and water, $V$ and $S$ are the volume and cross-sectional area of mitochondria, and $d$ is the thickness of a mitochondrion (~1 µm). Using values for the geometry and properties of a mitochondrion, the pressure induced by MitoCeption centrifugation was ~1.6 Pa.

### Numerical simulation of MitoPunch internal pressure

The finite element method (COMSOL Inc, Burlington, MA, Multiphysics 5.3) was used to simulate the pressure inside the MitoPunch PDMS chamber. We constructed the simulation geometry according to real device dimensions. Piston movement was applied as initial displacement in the y direction. Considering the incompressibility of the aqueous medium inside the PDMS chamber, the volume of the chamber was maintained constant while solving for the stress distribution of all the materials.

### SIMR clone isolation

Mitochondrial recipient and vehicle delivery control 143BTK– ρ0 cells were grown in complete medium supplemented with 50 mg/L uridine for 3 days following mitochondria or vehicle transfer. After 3 days, the medium was changed to SIMR selection medium (complete medium with 10% dialyzed FBS (Life Technologies, Carlsbad, CA, Cat. # 26400–044)) and medium was exchanged daily. After the vehicle delivery control sample died and clones emerged on mitochondrial transfer plates (~7 days SIMR selection medium), clones were isolated using cloning rings or plates were fixed and stained with crystal violet for counting.

Mitochondrial recipient and vehicle delivery control BJ ρ0 and B16 ρ0 cells were grown in complete media supplemented with 50 mg/L uridine for 3 days following mitochondria transfer. After 3 days, the medium was changed to SIMR selection medium and exchanged daily. On day 5 post-delivery, cells were shifted to galactose selection medium (glucose-free, galactose-containing

medium [DMEM without glucose, Gibco, Cat. # 11966025] supplemented with 10% dialyzed FBS and 4.5 g/L galactose [Sigma-Aldrich, Cat. #G5388-100G]). After the vehicle delivery control sample died and clones emerged on mitochondrial transfer plates (~36 hr in galactose selection medium), clones were isolated using cloning rings or plates were fixed and stained with crystal violet for counting.

## Crystal violet staining and clone counting

Media was aspirated from 10 cm plates before fixation with 1 mL freshly diluted 4% paraformaldehyde in 1× DPBS for 15 min at RT. Fixative was removed and 1 mL 0.5% w/v crystal violet solution (Thermo Fisher Scientific, Cat. # C581-25) dissolved in 20% methanol in water was applied to each plate and incubated for 30 min at RT. Crystal violet was removed and plates were washed 2× with deionized water before drying overnight at RT. Dried plates were photographed and crystal violet stained clones were counted manually using FIJI (*Schindelin et al., 2012*).

## Imaging flow cytometry

Mitochondria were transferred to recipient cells, which were harvested and collected in 1.5 mL tubes. Samples were centrifuged 5 min at 1000 × *g*, supernatant was aspirated, and cells were washed 3× with 0.5 mL 1× DPBS, pH 7.4. The DPBS was aspirated and cells were fixed in 100 µL freshly diluted 4% paraformaldehyde (Thermo Fisher Scientific, Cat. # 28906) for 15 min on ice. Fixative was diluted with 1 mL of 1× DPBS, pH 7.4, and 5% FBS and centrifuged for 10 min at 500 × *g*. Supernatant was removed and cells resuspended in 1× DPBS, pH 7.4, with 5% FBS. Imaging flow cytometry was performed using an ImageStream MarkII platform and analyzed using the IDEAS 6.2 software package (Luminex, Austin, TX).

## Confocal microscopy

$1 \times 10^5$ cells were plated in 6-well dishes with 2 mL of media on glass coverslips (Zeiss, Oberkochen, Germany, Cat. # 474030–9000) ~24 hr prior to sample preparation. Medium was aspirated and samples were fixed with 0.5 mL freshly diluted 4% paraformaldehyde in 1× DPBS, pH 7.4 pipetted onto samples and incubated for 15 min at RT. Paraformaldehyde was removed and samples were washed 3× with 5 min 1× DPBS incubations. Samples were then permeabilized by 10 min RT incubation in 0.1% Triton-X 100 (Sigma, St. Louis, MO, Cat. # X100). Permeabilized samples were washed 3× with 1× DPBS and then incubated for 1 hr at RT with 2% bovine serum albumin (BSA) dissolved in 1× DPBS blocking buffer. Blocking buffer was aspirated and cells incubated for 1 hr at RT with a 1:1000 dilution of primary antibodies in 2% BSA blocking buffer against dsDNA (Abcam, Cambridge, United Kingdom, Cat. # ab27156) and TOM20 protein (Abcam, Cat. # ab78547), and then washed 3× with 5 min 1× DPBS incubations. Cells were then incubated with secondary antibodies (Invitrogen, Cat. # A31573 and A21202) diluted 1:100 in 2% BSA blocking buffer protected from light for 1 hr at RT. After incubation with secondary antibodies, samples were washed 3× with 5 min 1× DPBS incubations and mounted on microscope slides.

To mount, samples were removed from the 6-well dish and rinsed by dipping in deionized water, dried with a Kimwipe, and mounted using ProLong Gold Antifade Mountant with DAPI (Invitrogen, Carlsbad, CA, Cat. # P3691) or ProLong Glass Antifade Mountant with NucBlue Stain (Thermo Fisher Scientific, Cat # P36985) on microscope slides (VWR, Cat. # 48311–601). Samples were dried at RT protected from light for 48 hr prior to confocal imaging with a Leica SP8 confocal microscope (Leica, Wetzlar, Germany) and later analyzed with either LAS X Lite 3.7.1.21655 (Leica) for two-dimensional image preparation or Imaris File Converter 9.5.1 (Oxford Instruments, Abingdon, United Kingdom) and Imaris Viewer 9.5.1 (Oxford Instruments) for Z-stack analysis.

To perform confocal imaging on cells immediately following mitochondrial transfer, $1 \times 10^5$ cells were plated in 6-well dishes with 2 mL of media on glass coverslips for coincubation and MitoCeption or seeded onto 12-well filter inserts as described above for MitoPunch ~24 hr prior to delivery. Immediately prior to mitochondrial transfer, coincubation and MitoCeption samples were stained with 1× CellMask Green PM (Molecular Probes, Eugene, OR, Cat. # C37608) diluted in warm medium for 10 min and washed twice in DPBS, and MitoPunch samples were stained with 5 µg/mL Alexa Fluor 488 conjugated Wheat Germ Agglutinin (Invitrogen, Cat. # W11261) diluted in warm media for 10 min and washed twice in DPBS. Following delivery, culture medium was removed and 1

mL freshly diluted 4% paraformaldehyde in 1× DPBS, pH 7.4, was pipetted onto samples and incubated for 15 min at RT. Paraformaldehyde was aspirated and samples were washed 3× with 1× DPBS, pH 7.4. Samples were further washed with DPBS 3× with 5 min RT incubation per wash. Mito-Punch filters were removed from the plastic insert using an inverted P1000 pipette tip. Samples were mounted and imaged as described above.

### OCR measurements

OCR measurements were performed using a Seahorse XFe96 Extracellular Flux Analyzer (Agilent, Santa Clara, CA). $2 \times 10^4$ cells were seeded into each well of a V3 96-well plate (Agilent, Cat. # 101085–004) and cultured 24 hr before measuring OCR. The Agilent Seahorse mitochondrial stress test was used to quantify OCR for basal respiration and respiration following the sequential addition of the mitochondrial inhibitors oligomycin, carbonyl cyanide-p-trifluoromethoxyphenylhydrazone (FCCP), and antimycin A. Data were analyzed using the Wave 2.6.2 software package (Agilent).

### PI staining, delivery, and flow cytometry

Cells ($1 \times 10^5$) were plated for delivery and incubated overnight. Media was changed to FluorBrite DMEM media (ThermoFisher Scientific, Cat. # A1896701) with 3 µM PI (Thermo Fisher Scientific, Cat. # P1304MP) immediately before transfer. MitoCeption and coincubation were carried out as described above, and MitoPunch was performed with PI FluorBrite medium loaded into the PDMS reservoir and incubated for 15 min at 37°C and 5% $CO_2$. All samples were washed with 1× DPBS and collected using Accutase. Samples were collected in flow cytometry tubes and centrifuged 5 min at $500 \times g$. Samples were washed with 1× DPBS with 5% FBS three times and analyzed on a BD Fortessa flow cytometer (BD Biosciences, San Jose, CA) and data were processed using FlowJo 10.6.2 (BD Biosciences).

### Quantification and statistical analysis

All information pertaining to experimental replication are found in the figure legends. Mitochondrial transfer experiments were performed in technical triplicate to enable calculation of standard deviation unless otherwise indicated, and oxygen consumption measurements were collected in technical quadruplicate or quintuplicate indicated in the legend of *Figure 4*. Investigators were blinded for SIMR colony counting analysis. All column heights represent the mean of technical triplicate results unless noted otherwise. All error bars in this manuscript represent standard deviation of three technical replicates unless otherwise specified in the figure legend.

## Acknowledgements

We thank Rebeca Acin-Perez, Linsey Stiles, and Orian Shirihai of the UCLA Metabolomics Core for help with Seahorse XF Analyzer assays. We thank Zoran Galic, Alejandro Garcia, and Salem Haile of the UCLA Jonsson Comprehensive Cancer Center Flow Cytometry Core Laboratory and Felecia Codrea, Jessica Scholes, and Jeffrey Calimlim of the UCLA Broad Stem Cell Research Center Flow Cytometry Core for assistance with cellular analysis. We thank Laurent Bentolila and Matthew J Schibler of the UCLA Advanced Light Microscopy/Spectroscopy core for assistance with confocal microscopy. We also thank Emma Dawson, Lynnea Waters, Natasha Carlson, and Christopher Sercel for critical feedback and assistance with this manuscript.

## Additional information

### Competing interests

Ting-Hsiang Wu: T.-H.W. was an employee of NanoCav, LLC, and is currently employed by NantBio, Inc and ImmunityBio, Inc. Shahrooz Rabizadeh: S.R. is a board member of NanoCav, LLC, and employed by NantBio, Inc, ImmunityBio, Inc, and NantOmics, LLC. Kayvan R Niazi: K.R.N. is a board member of NanoCav, LLC, and employed by NantBio, Inc and ImmunityBio, Inc. Pei-Yu Chiou: P.-Y. C. is a co-founder, board member, shareholder, and consultant for NanoCav, LLC, a private start-up company working on mitochondrial transfer techniques and applications. Michael A Teitell: M.A.T. is

a co-founder, board member, shareholder, and consultant for NanoCav, LLC, a private start-up company working on mitochondrial transfer techniques and applications. The other authors declare that no competing interests exist.

## Funding

| Funder | Grant reference number | Author |
|---|---|---|
| National Institutes of Health | T32CA009120 | Alexander J Sercel Alexander N Patananan |
| National Institutes of Health | T32GM007185 | Alexander J Sercel |
| American Heart Association | 18POST34080342 | Alexander N Patananan |
| National Institutes of Health | T32GM008042 | Amy K Yu |
| National Science Foundation | CBET 1404080 | Pei-Yu Chiou |
| National Institutes of Health | R01GM114188 | Pei-Yu Chiou Michael A Teitell |
| Air Force Office of Scientific Research | FA9550-15-1-0406 | Pei-Yu Chiou Michael A Teitell |
| National Institutes of Health | R01GM073981 | Michael A Teitell |
| National Institutes of Health | R01CA185189 | Michael A Teitell |
| National Institutes of Health | R21CA227480 | Michael A Teitell |
| National Institutes of Health | P30CA016042 | Michael A Teitell |
| CIRM | RT3-07678 | Michael A Teitell |

The funders had no role in study design, data collection and interpretation, or the decision to submit the work for publication.

## Author contributions

Alexander J Sercel, Conceptualization, Data curation, Formal analysis, Funding acquisition, Validation, Investigation, Visualization, Methodology, Writing - original draft, Project administration; Alexander N Patananan, Conceptualization, Funding acquisition, Validation, Investigation, Methodology, Writing - review and editing; Tianxing Man, Software, Formal analysis, Investigation, Methodology; Ting-Hsiang Wu, Software, Methodology; Amy K Yu, Formal analysis, Investigation; Garret W Guyot, Investigation; Shahrooz Rabizadeh, Kayvan R Niazi, Funding acquisition; Pei-Yu Chiou, Software, Funding acquisition, Methodology; Michael A Teitell, Conceptualization, Supervision, Funding acquisition, Methodology, Writing - review and editing

## Author ORCIDs

Alexander J Sercel ⓘD https://orcid.org/0000-0002-0749-2162
Alexander N Patananan ⓘD https://orcid.org/0000-0001-9458-9968
Michael A Teitell ⓘD https://orcid.org/0000-0002-4495-8750

## Decision letter and Author response

Decision letter https://doi.org/10.7554/eLife.63102.sa1
Author response https://doi.org/10.7554/eLife.63102.sa2

# Additional files

## Supplementary files

• Transparent reporting form

## Data availability

Figure 1-source data 1. Numerical simulation of MitoPunch pressure generation during mitochondrial delivery. Cited in the legend of Figure 1.

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
