## [Decision Letter]

**Acceptance summary:**

The manuscript details novel methodology for introducing functional mitochondria into mammalian cells, which work in both immortalized and primary cell lines. In addition, the authors provide data demonstrating a failure of mitochondrial transfer via simple co-incubation with mitochondria, an important observation. Overall, this is a promising new technique, and should be of high interest to researchers studying mitochondrial-nuclear crosstalk, mitonuclear compatibility, and related fields.

**Decision letter after peer review:**

Thank you for submitting your article "Stable transplantation of human mitochondrial DNA by high-throughput, pressurized mitochondrial delivery" for consideration by *eLife*. Your article has been reviewed by two peer reviewers, one of whom is a member of our Board of Reviewing Editors, and the evaluation has been overseen by Matt Kaeberlein as the Senior Editor. The following individual involved in review of your submission has agreed to reveal their identity: Scott R Kennedy (Reviewer #2).

The reviewers have discussed the reviews with one another and the Reviewing Editor has drafted this decision to help you prepare a revised submission.

Summary:

Research into certain aspects of mitochondrial biology has been limited by a lack of robust methodology for introducing isolated mitochondria into recipient cells. To date, well-validated mitochondrial transfer methods have been highly specialized and extremely low throughput in nature. Sercel et al. describe a novel mitochondrial transfer technology that uses a plunger actuated mechanism to ballistically deliver isolated mitochondria into mtDNA-deficient recipient cells. They report evidence that transferred mitochondria are stably integrated into the recipient cell and compare this new approach to the previously described MitoCeption technology. The technique works in both immortalized and primary cell lines, with mitochondria persisting in numerous clones derived from individual cells. Furthermore, the authors provide data demonstrating a failure of mitochondrial transfer via simple co-incubation, an important observation.

Overall, this is an exciting new technique, and should be of high interest to researchers who study mitochondrial disease or the basic biology of mitochondria. However, there are a number of issues that should be addressed prior to publication.

1) Methodologic and technical details are insufficient in the current manuscript. We remind the authors that for this type of manuscript, "The article should fully describe the tool or resource so that prospective users have all the information needed to deploy it within their own work… methodological advances need to be comprehensibly described, along with details of the reagents and equipment, and their sources." Please provide detail, including improved diagrams and instructions, sufficient so that independent will be able to reproduce the approach. Where values may need to be empirically determined by other laboratories (such as how to determine the ideal voltage) details should be provided in how to approach the issue. In addition, authors need to note the approximate passage number/population doubling of the primary cells used and verify that they were similar between comparisons, as onset of senescence would undoubtedly have a dramatic impact of success in these assays. Figure legend and text also need more detail – for example, in Figure 4, it can be assumed that immune staining was used for TOM20 detection, and in section “ SIMR cells rescue ρ0 mitochondria network morphology and respiration” needs to state how mtDNA was imaged. Readers should not have to reference methods for these simple details.

2) Experiments showing stable incorporation of functional mitochondria were performed in uridine-free media to select for cells with mitochondria. Authors should demonstrate that mitochondria remain following passage in uridine-containing media. This data may in fact be present in the freeze-thaw culture experiments, but needs to be explicitly stated – it is not clear which media was used in those assays.

3) Additional information is needed regarding the mitochondrial isolation methodology, and data showing variability between mitochondrial preparations is necessary, rather than just technical replicates from the same isolation. It is impossible to know if the preparation used is an outlier in terms of performance, and it is important to know how reproducible the entire technique is. In regard to the technique, providing the kit used is simply not sufficient, particularly given that the brief description provided does not seem to match with the kit protocol available online. Further, one reviewer expressed concern that without these details we cannot be confident that contaminating donor cells weren't present in the isolated mitochondria, which may survive and outcompete mitochondrial deficient cells. It would also be useful to indicate, at least in the Discussion, what impact different methods of mitochondrial isolation are expected to have on success in these procedures.

4) The first data section, “ Mitochondrial delivery into transformed and primary cells”, should be re-worded. The data does not assess delivery INTO cells, as suggested by the prose; rather, the data presents co-association after 2 hours. The subsequent data demonstrates that actual delivery into cells does not correlate with the co-staining shown in this section, and the SEM data indicates that co-incubation simply results in an extracellular coating of mitochondria. The subsequent data clearly show a lack of actual transfer using co-incubation. I believe this is what the authors intended.

5) There are a few issues with Figure 3. In the titration experiments, the interpretation is not supported. The authors show that co-staining by flow cytometry does not mean mitochondria were taken into recipient cells, so in Figure 3G the right half of the panel is not useful and the 4-fold increase in efficiency is a misrepresentation. The data currently shown is “SIMR colonies per 104 cells that show co-staining with ds-RED”, not “recipient cells”. It is not clear that MitoPunch actually outperforms MitoCeption. The left half of the panel would suggest they perform similarly in this cell type. Truthfully, given the distinct properties of these two methods, it may not be reasonable to try to directly compare their relative efficacy in such a manner. It is sufficient to show they both work well and in a mitochondrial-mass dose-dependent manner, whereas co-incubation fails at all mitochondrial mass concentrations. Given the current state of the field, the new technique is still exciting, even if it has not been tested alongside MitoCeption in a way which allows direct head to head comparisons of efficacy. Future studies may delve deeper into the cell type, conditions, etc, where each technique best performs, but if the authors wish to claim here that MitoPunch is superior, additional data will be needed. It is assumed by the reviewer, but needs to be clarified/confirmed, that in the Figure 3G experiments the same cell number was used.

---

## [Author Response]

Revisions for this paper:1) Methodologic and technical details are insufficient in the current manuscript. We remind the authors that for this type of manuscript, "The article should fully describe the tool or resource so that prospective users have all the information needed to deploy it within their own work… methodological advances need to be comprehensibly described, along with details of the reagents and equipment, and their sources." Please provide detail, including improved diagrams and instructions, sufficient so that independent will be able to reproduce the approach. Where values may need to be empirically determined by other laboratories (such as how to determine the ideal voltage) details should be provided in how to approach the issue.

We included a new Figure 1—figure supplement 1 showing an annotated photograph of the MitoPunch device and also included new subsections in the Materials and methods detailing how to construct, tune, and operate the device as suggested.

In addition, authors need to note the approximate passage number/population doubling of the primary cells used and verify that they were similar between comparisons, as onset of senescence would undoubtedly have a dramatic impact of success in these assays.

We included these details in the revised text to note the approximate passage number of the BJ ρ0 cells used in the study, as suggested.

Figure legend and text also need more detail – for example, in Figure 4, it can be assumed that immune staining was used for TOM20 detection, and in section “ SIMR cells rescue ρ0 mitochondria network morphology and respiration” needs to state how mtDNA was imaged. Readers should not have to reference methods for these simple details.

We updated the figure legends and revised the Results text to include greater procedural details throughout the manuscript, including particular attention to our discussion of immunostaining, to clarify methodological details, as suggested.

2) Experiments showing stable incorporation of functional mitochondria were performed in uridine-free media to select for cells with mitochondria. Authors should demonstrate that mitochondria remain following passage in uridine-containing media. This data may in fact be present in the freeze-thaw culture experiments, but needs to be explicitly stated – it is not clear which media was used in those assays.

We indicated in the revised text that the SIMR clones were grown in uridine supplemented medium following their initial derivation as suggested.

3) Additional information is needed regarding the mitochondrial isolation methodology, and data showing variability between mitochondrial preparations is necessary, rather than just technical replicates from the same isolation. It is impossible to know if the preparation used is an outlier in terms of performance, and it is important to know how reproducible the entire technique is.

We performed additional experiments to demonstrate the reproducibility of the mitochondrial isolation method and the MitoPunch technique, as suggested. In short, we performed three triplicate MitoPunch experiments using three independent mitochondrial isolations and measured clone formation and mitochondrial mass from each preparation. We included this additional data as Figure 3—figure supplements 3 and 4 and also included relevant revised text in the Results section as suggested.

In regard to the technique, providing the kit used is simply not sufficient, particularly given that the brief description provided does not seem to match with the kit protocol available online. Further, one reviewer expressed concern that without these details we cannot be confident that contaminating donor cells weren't present in the isolated mitochondria, which may survive and outcompete mitochondrial deficient cells.

We provided additional details and clarification in the revised Materials and methods text regarding our mitochondrial isolation method, clearly describing the exact procedure used. We also performed an additional experiment represented in Figure 3—figure supplement 1 to demonstrate the lack of surviving donor cells in our mitochondrial preparations.

It would also be useful to indicate, at least in the Discussion, what impact different methods of mitochondrial isolation are expected to have on success in these procedures.

We developed the MitoPunch device and optimized operational procedures using mitochondria isolated by Dounce homogenization with standard mitochondrial isolation buffers (data not shown), and later adopted needle disruption and the Qiagen Qproteome kit to streamline the pipeline and improve reproducibility. We have not used other methods to isolate mitochondria for mitochondrial transfer, but we speculate that preparations generated in other ways would also yield SIMR clones. We included revised text in the Discussion section of the manuscript to address this point.

4) The first data section, “ Mitochondrial delivery into transformed and primary cells”, should be re-worded. The data does not assess delivery INTO cells, as suggested by the prose; rather, the data presents co-association after 2 hours. The subsequent data demonstrates that actual delivery into cells does not correlate with the co-staining shown in this section, and the SEM data indicates that co-incubation simply results in an extracellular coating of mitochondria. The subsequent data clearly show a lack of actual transfer using co-incubation. I believe this is what the authors intended.

We thank the reviewers for noting this discrepancy. We intended the prose to indicate the occurrence of mitochondrial association with recipient cells following delivery without an implication of the mechanism of co-localization. We revised the text to better represent the result and reflect our intended conclusion, as suggested.

5) There are a few issues with Figure 3. In the titration experiments, the interpretation is not supported. The authors show that co-staining by flow cytometry does not mean mitochondria were taken into recipient cells, so in Figure 3G the right half of the panel is not useful and the 4-fold increase in efficiency is a misrepresentation. The data currently shown is “SIMR colonies per 104 cells that show co-staining with ds-RED”, not “recipient cells”. It is not clear that MitoPunch actually outperforms MitoCeption. The left half of the panel would suggest they perform similarly in this cell type.

We agree with the reviewers that this result can lead to an unintended interpretation. We intended to show that the SIMR generation efficiency is greater for each cell that interacts with dsRed mitochondrial material by MitoPunch. To avoid confusion for readers, we removed this plot from the manuscript and revised the Results text.

Truthfully, given the distinct properties of these two methods, it may not be reasonable to try to directly compare their relative efficacy in such a manner. It is sufficient to show they both work well and in a mitochondrial-mass dose-dependent manner, whereas co-incubation fails at all mitochondrial mass concentrations. Given the current state of the field, the new technique is still exciting, even if it has not been tested alongside MitoCeption in a way which allows direct head to head comparisons of efficacy. Future studies may delve deeper into the cell type, conditions, etc, where each technique best performs, but if the authors wish to claim here that MitoPunch is superior, additional data will be needed. It is assumed by the reviewer, but needs to be clarified/confirmed, that in the Figure 3G experiments the same cell number was used.

Our intent was not to claim that MitoPunch is necessarily superior to MitoCeption, but to demonstrate that application of pressure during mitochondrial transfer enhances SIMR generation in different systems and also to highlight that the two methods have distinct use cases depending on the recipient cell type of interest. We included new text in the Discussion to clarify this point. We additionally clarified that the same number of cells were seeded for each replicate in Figure 3—figure supplement 5.